# The Effects of Biomass Combustion Ash and Lignin on the Properties of Cement Mortars and Their Environmental Impact

**DOI:** 10.3390/ma18092086

**Published:** 2025-05-02

**Authors:** Iwona Ryłko, Łukasz Bobak, Paweł Telega, Andrzej Białowiec

**Affiliations:** Department of Applied Bioeconomy, Wrocław University of Environmental and Life Sciences, 37a Chełmońskiego Str., 51-630 Wrocław, Poland; iwona.rylko-polak@upwr.edu.pl (I.R.); lukasz.bobak@upwr.edu.pl (Ł.B.); pawel.telega@upwr.edu.pl (P.T.)

**Keywords:** biomass combustion ash, waste lignin, sustainable construction, pollutant leachability, cement mortars

## Abstract

Combustion and hydrolysis of lignocellulosic biomass generate renewable energy and biofuels, but also yield by-products, such as biomass combustion ash (BCA) and waste lignin (WL). This study investigates the reuse of these by-products in cement mortars, promoting circular economy principles and sustainable construction practices. The addition of BCA at 1–10% improved mortar consistency, homogeneity, and adhesion—most notably, formulations with 5–10% BCA increased adhesion to EPS by up to 4.3%, and compressive strength remained above the 20 MPa threshold. WL additions of 0.5–1% enhanced viscosity and adhesion to both mineral and EPS substrates, with a 0.2% WL dosage improving adhesion to EPS by 9.4% compared to the control sample. Life Cycle Assessment (LCA) confirmed a reduction in the carbon footprint by up to 14% (from 1509.5 to 1297.5 Mg CO_2_/year), while VOC emissions remained within acceptable limits. Leachability tests confirmed safe environmental performance. The results validate BCA and WL as functional and eco-efficient additives in cementitious composites suitable for thermal retrofitting.

## 1. Introduction

Agricultural biomass plays a key role in the European Union’s energy transition, accounting for approximately 60% of the region’s renewable energy sources [1]. In the heating and cooling sector, this share increases to nearly 75% [2]. While the use of biomass supports energy diversification, job creation, and the reduction of greenhouse gas emissions, the combustion and hydrolysis of lignocellulosic materials result in poorly managed by-products [3]. The most prominent of these are biomass combustion ash (BCA) and waste lignin (WL), which are generated in significant quantities by the paper industry and in the production of second-generation bioethanol. Under Polish conditions—where biomass is abundantly available and the use of forest biomass is increasingly restricted—the sustainable management of these residues poses both environmental and technological challenges [4].

In recent decades, a wide range of industrial waste materials have been explored as additives in cement mortars, including coal fly ash, silica fume, metakaolin, and waste from glass and ceramics industries [4,5]. Many of these additives are known to improve the mechanical strength, durability, and rheological properties of cementitious composites [5]. For example, metakaolin enhances early-age strength, while silica fume improves resistance to aggressive environments. However, these conventional materials also have limitations—untreated coal fly ash can increase water demand or release undesirable volatiles, while other materials may suffer from inconsistent supply, high costs, or region-specific availability [6].

In contrast, BCA and WL represent renewable, locally sourced, and underutilized alternatives. BCA is derived from the combustion of biomass, rather than fossil fuels, which reduces its carbon footprint and environmental impact [7]. WL, a by-product of lignocellulosic biomass processing, serves as a natural plasticizer and can partially replace synthetic admixtures [8]. Despite these benefits, research into the performance of BCA and WL in cementitious applications remains limited, making them novel subjects of investigation in the context of circular construction materials.

This study aims to address this gap by evaluating the influence of BCA and WL as sustainable additives to cement mortars. The research focuses on their effects on rheological behavior, mechanical properties, and environmental aspects, such as VOC emissions, pollutant leachability, and carbon footprint (LCA). Special attention is given to formulations intended for use in thermal insulation systems, including adhesion to mineral and polystyrene (EPS) substrates.

The novelty of this work lies in the synergistic use of two distinct industrial by-products—fluidized bed boiler ash and sulfonated lignin—in the development of adhesive mortars for sustainable construction. The results indicate that, when used in optimized proportions (BCA ≤ 10%, WL = 0.5–1%), the modified mortars not only meet technical standards but also reduce VOC and CO_2_ emissions. This approach contributes to circular economy strategies and offers a practical solution for reducing the environmental footprint of construction materials.

## 2. Materials and Methods

### 2.1. Materials

In the adhesive formulations developed [9], the main ingredient is Portland cement CEM II 42.5 A-V, accounting for approximately 30% of the composition in all formulations. This cement belongs to the CEM II class, which is characterized by lower CO_2_ emissions during the production process compared to traditional Portland cement. The choice of this type of cement is in line with the new regulations of the European Union, which promote the use of low-emission building materials [10]. In Poland, the switch to CEM II-grade cement is becoming standard, driven by increasing environmental requirements and the drive to reduce the carbon footprint of the construction industry [11].

Further components are quartz sand (0.1–0.5 mm), which acts as a filler material, the proportion of which decreases in subsequent recipe variants [9]; cellulose ether [9], a long-molecular hydroxypropyl methylcellulose with a viscosity of approximately 45.000 mPa-s, used in an amount of approximately 0.3% to stabilize the mixture; and a re-dispersible polymer based on ethylene and vinyl acetate, added in an amount of 1%, which acts as a film-forming agent to improve adhesion, deformability, and resistance to cracking in polymer-modified cementitious adhesives [9].

The supplements used are as follows:BCA waste, from bottom sand, with the code 10 01 01, generated in a fluidized bed boiler from the combustion of 100% biomass, without coal dust [12]. This partially replaces quartz sand, in a proportion ranging from 5% in the first formula to 20% in the fourth formula. Waste material obtained from Enea Elektrownia Połaniec S.A., 28-230 Połaniec, Poland.WL is produced by the sulfite digestion process, which involves treating woodchips with sulfur dioxide. The woodchips are first boiled in a mixture of water, sulfur dioxide, and calcium hydroxide to break down the lignin. The resulting pulp is then washed, and sodium lignosulphonate is extracted by filtering and evaporating the spent sulfite lye [13,14]. In building materials, WL acts as a plasticizer and water regulator, improving the workability, plasticity, and flexibility of the mixture, which increases the strength of the material. Its content ranges from 0 to 4% [9]. Waste material obtained from Proexport Chemicals, importer of concrete admixtures.

### 2.2. Methods

#### 2.2.1. Configuration of Mortar Formulations

As part of the research, a simple adhesive recipe was prepared for the fixing of polystyrene and mineral wool, which is a key component in insulation systems (Table 1 and Table 2). The formulations were developed following the principles of mixed building design, based on experience in the field and the available technical literature [9]. Table 1 contains formulas with added BCA, Table 2 contains formulas with added WL. Efforts were made to ensure compliance with the standard requirements for adhesive mortars, which guarantee high quality and thermal efficiency of the product.

#### 2.2.2. Preparation Process

All ingredients were measured according to the formulations. The dry components were first homogenized for 30 s. Subsequently, water was added gradually while mixing with a mechanical stirrer at 600 rpm for 90 s, (standard mixer, designed for mixing cement mortars in planetary mixing mode). After a rest period of 2 min, mixing was resumed for an additional 60 s to ensure a uniform, lump-free mixture [15]. The consistency was evaluated using the flow table method, following EN 1015-3 [15]. The target spread diameter was set at 140 ± 5 mm, appropriate for cementitious adhesives used in ETICS applications. This target was used for all formulations to ensure comparable rheological behavior [15].

The selection of dosage ranges for biomass combustion ash (BCA) and waste lignin (WL) was informed by both preliminary laboratory tests and the existing literature [7,8]. The BCA content was varied between 5% and 20% by weight (Table 1), as previous studies [4] have demonstrated its effectiveness as a pozzolanic filler in quantities up to approximately 15–20%, beyond which workability and adhesion may significantly deteriorate due to increased water demand and porosity. In the case of sodium lignosulphonate (WL), a range of 0–4% by weight was selected (Table 2), based on its known plasticizing properties and its potential impact on VOC emissions [8]. Earlier research [3] and our preliminary testing suggested that exceeding 2.5–3% WL could result in reduced mechanical performance and excessive softening. These ranges were therefore chosen to explore both technical performance and environmental impact within practical limits suitable for building applications. Laboratory tests were then carried out to evaluate parameters such as workability, water retention, adhesion, and compressive strength. The research also included testing of a starter formulation and further variants enriched with BCA and WL, allowing for a thorough analysis of their properties and suitability in insulation systems.

#### 2.2.3. Oil Number Test

An oil number test was carried out for each sample. The oil number is the amount of oil (usually linseed or another organic solvent) needed to produce a paste of the correct consistency from a given amount of powder material, such as cement, fly ash, quartz sand, or another mineral material. This is an important parameter in the evaluation of materials used in construction, because it provides an estimate of the number of organic substances (such as plasticizers) that will be needed to bind the material particles [16].

A specified amount of material, e.g., 100 g of quartz sand, cement, or mixture, was placed on the pan of a balance. The solvent (water) was gradually added to each sample in small amounts, stirring constantly with a spatula or mortar. The process of adding the solvent was continued until the resulting mixture had a paste-like consistency—that is, all particles were evenly coated with oil, and the mixture became plastic but not overly sticky. The oil number and amount of solvent (in grams) needed per 100 g of dry material were measured [16]. The results are included in Table 3.

#### 2.2.4. Testing of Mortar Properties

For each of the samples, detailed tests were carried out to assess the effects of the addition of ash and lignin on the physical and mechanical parameters of the mortar, based on the EAD (European Assessment Document) standard 040083-00-0404 [17]. This study employed European technical assessment methods for construction materials, including cement-based mixtures and products used in renovation and finishing work.
Consistency determination: The Navikow fall cone method is a test conducted with an apparatus designed to determine the consistency of construction mortars, following EN-12350-2 [18]. The principle of the test is to determine the depth of immersion of the measuring cone in the mortar, measured in centimeters along the cone’s surface.Adhesion to substrate: This test aims to assess the material’s ability to adhere to different substrates, such as concrete. This is crucial for repair and finishing products. The tests were carried out on polystyrene (EPS) board substrates and standard concrete substrates, complying with EAD 040083-00-0404, Section 2.2.11.2. and Section 2.2.11.3 [17]. The tests were carried out on EPS Thermo Fasada Extra 038 (TR 100), Arsanit producer polystyrene boards, using the formulations developed, conditioned for 28 days under laboratory conditions of 23 °C and ±50%HR.Bending and shear strength: These tests measure the mechanical properties of a material, i.e., its ability to withstand bending and compressive loads. This is important to ensure adequate load-bearing capacity and stability of a structure. The tests were carried out according to EN 998-1:2016-12 [19], on conditioned beams under laboratory conditions of 23 °C and ±50%HR, with the beams subjected to bending and compression using a test press.

#### 2.2.5. Leachability of Contaminants from Building Materials—Relevance of Leachability Testing

Investigating the leachability of contaminants from building materials is extremely important both for manufacturers and for environmental protection. For manufacturers, compliance with regulations is crucial. Building materials must meet certain environmental and technical standards, including leachability limits for harmful substances [20]. The reputation and quality of products an important factor. Using test results to verify that products are safe and durable can influence customer confidence. Knowledge of leachability allows formulations to be modified to reduce the risk of emissions.

For the environment, reducing pollution of soil, water bodies, and aquifers is paramount. Construction materials can be a source of leachable pollutants that affect water quality. The use of industrial waste, such as BCA and WL, requires an assessment of its environmental impact to prevent potential side effects.

The Regulation of the Minister of the Economy of 16 July 2015 defines the criteria for accepting inert waste at landfills, and its Annex 2 [20] specifies permissible pollutant values and test methods to assess their environmental impact. The performance of laboratory tests to confirm that standards for chemical composition, heavy metal content, and leaching are met is a key aspect.

These tests are carried out following EN 12457-4:2006 [21], which defines the requirements for testing laboratories, ensuring the reliability of the results. The process involves taking representative samples of waste, preparing them, chemically analyzing them, and evaluating the results for compliance with acceptable standards. Thanks to these regulations, waste can be stored in an environmentally safe manner, minimizing the risk of water and soil contamination.

#### 2.2.6. Volatile Compound Emissions and Their Significance

Volatile organic compounds (VOCs) are a group of chemicals that can readily change to a gaseous state at room temperature, due to their high vapor pressure. In the context of cement mortars and other building materials, VOC emissions play an important role in affecting air quality, human health, and the environment [22]. Sources of VOCs in cement mortars include chemical admixtures such as plasticizers and superplasticizers, setting retarders, hydrophobic agents, and dyes. Organic materials as a BCA and lignin are resources that can be used to replace these. In addition, cement hydration processes can generate VOCs as by-products.

Regulations are reducing VOC emissions and promoting sustainable building materials. Products with low VOC emissions are required by green building standards such as LEED, BREEAM, or WELL. It is only possible to obtain environmental certification (e.g., EU Ecolabel) if strict emission standards are met [22].

The gas chromatography (GC) [23,24] method was chosen for the prepared samples; it is one of the most commonly used methods for investigating VOCs in cement mortars, especially when combined with mass spectrometry (GC-MS). The chosen method is characterized by high sensitivity and precision, enabling the identification of trace amounts of compounds [25].

The analysis of the samples was conducted using an Agilent GC-MS system (GC 7890B/MS 5977B), produced Agilent Technologies Inc., CA, USA, equipped with a DB-5MS column (30 m × 0.25 mm × 0.25 μm; AGILENT). The analytes were adsorbed using a FIB-C-WR-95/10-P1 Carbon WR/PDMS fiber (PALSystem) with a fiber thickness of 95 µm and a fiber length of 10 mm, installed in the arm of an MPS Robotic multi-functional autosampler (Gerstel), producers GERSTEL GmbH & Co. KG, Germany. In the analysis, 2.5 ppm of Cariophilene was added as an internal standard. The fiber was preconditioned by heating at 270 °C for 30 min, and for 1 min before and after each analysis under the same conditions. Before extraction, the sample was conditioned at 60 °C for 10 min, followed by volatile compound extraction at the same temperature for 10 min. Desorption of the analytes was carried out in the injector for 5 min [23,24].

The instrument operating conditions were as follows: an injector temperature of 250 °C; a split ratio of 10:1; and a helium flow rate of 1.0 mL·min^−1^. The temperature program was as follows: a 40 °C hold time for 2 min, then an increase to 150 °C at a rate of 5 °C·min^−1^, followed by an increase to 250 °C at a rate of 10 °C·min^−1^. The quadrupole, ion source, and transfer line temperatures were set at 150 °C, 230 °C, and 250 °C, respectively. Identification of volatile organic compounds was performed using the NIST17 [25] mass spectral library.

#### 2.2.7. Carbon Footprint Estimated Using Life Cycle Assessment (LCA) Software

The analysis aimed to determine the annual carbon footprint (CO_2_ emissions) for a reference sample and recipe variants containing waste raw materials (biochar, ash, sodium lignosulphonate). The calculations were performed on a ‘cradle-to-gate’ basis, i.e., they included the emissions associated with the production and delivery of raw materials to the plant.

All formulation variants contained the same proportion of cement (30%). The differences were in the amount of sand and the proportions of ash, lignosulphonate, and polymer additives (cellulose ether and polymer). Emission values taken from Ecoinvent’s EPD data were used for the calculations. For each formulation, the following calculation was used:

The proportion of each ingredient in the formulation was multiplied by its emission factor. The emissions of all components for 100 kg of the mixture were summed. The annual emissions were calculated as follows:$$Carbon\;footprint\;[MG\frac{{CO}_{2}}{year}=\frac{emission\;per\;100\;kg\times total\;annual\;mass\;[kg]}{100\times 1000}$$

Based on the carbon footprint calculation, a Life Cycle Assessment was also performed, following ISO 14040/14044 [26] and ISO 14067:2018 [27], for the production of the formulations developed (‘cradle-to-gate’ analysis). The analysis (Module A1–A3) included the following:The production and supply of raw materials (Module A1);The mixing process (optional addition of energy consumption) (Module A2);Emissions resulting from the production of materials (Module A3).

A Life Cycle Assessment of each product was performed the calculate, among other things, the carbon footprint of the product, including production, transport, and operation [26,27,28]; and the consumption of raw materials and energy and their impacts on the energy efficiency of materials and on the environment, such as greenhouse gas emissions, water consumption, or the potential toxicity of materials.

## 3. Mechanical and Technological Performance of Cement Mortars

### 3.1. Determination of the Oil Number

Figure 1 illustrates the effect of BCA and WL content on the oil number. The observed increase in the oil number with rising BCA and WL content indicates greater water demand, which may present challenges in practical construction settings. Elevated water demand can lead to reduced workability, longer mixing times, and potential difficulties in achieving uniform dispersion, particularly in on-site applications where controlled conditions are limited. Moreover, higher water content may negatively affect early-age strength and extend curing durations, especially in formulations with >15% BCA. To mitigate these effects, the use of superplasticizers or water-reducing admixtures could be considered to improve flow characteristics without compromising mechanical performance. Alternatively, optimized mixing sequences, including pre-soaking or staged water addition, may enhance paste homogeneity. These practical considerations are crucial for translating laboratory findings into viable building applications, especially in energy retrofit and insulation systems, where application consistency is key [16].

### 3.2. Mortar Properties

#### 3.2.1. Water Absorption Capacity—BCA and Oil Number

The oil number test assesses a material’s ability to adsorb liquids, which has a direct bearing on the water requirements of cementitious mixtures. The results show that as the BCA (fly ash) content increases, the oil number increases, suggesting a higher specific surface area and greater adsorption capacity (Table 3, water–cement ratio).

**Table 3 materials-18-02086-t003:** Averaged test results for formulations developed with the addition of BCA content. Bolded numbers show which formulations meet the standards for mortar.

Ingredient	F0	F1	F2	F3	F4	Requirement [17,19]
**Water–cement ratio** (water to cement ratio by oil number)	0.21	0.22	0.24	0.25	0.26	
**Consistency** **(Navikow)**	**6.5**	**7**	**7**	**7**	**7**	6.5–7.5
**Adhesion to EPS**	**0.11**	**0.11**	**0.09**	0.07	0.06	≥0.08 MPa
**Adhesion to concrete substrate**	**0.3**	**0.28**	**0.25**	0.15	0.12	≥0.25 MPa
**Adhesion to EPS at +5 °C (EPS)**	**0.11**	**0.11**	**0.09**	0.07	0.06	≥0.08 MPa
**Adhesion to concrete substrate +5 °C (EPS)**	**0.29**	**0.28**	**0.25**	0.13	0.1	≥0.25 MPa
**Flexural strength**	6.4	5.5	5.5	5.2	4.6	Declared value
**Compressive strength**	**25.5**	**24.6**	**23.0**	**21.8**	**20.5**	(≥20 MPa) EN 1015-11:2001+A1:2007

BCA, as a fine material with a high specific surface area, absorbs larger amounts of water, leading to an increase in batch water requirements in concrete mixtures. At low doses, the ash can act as a filler, improving the workability and fluidity of the mix, sometimes even resulting in a reduction in the amount of water required. However, at higher BCA contents, the material starts to act as a strong water adsorbent, resulting in more water being retained in the microstructure of the ash particles, thus requiring the addition of more water to achieve the desired mix consistency [29].

The observation of changes in oil number as a function of BCA content confirms that the higher the specific surface area and adsorption capacity, the higher the water requirement, which is crucial when designing the optimum proportions in cementitious mixtures [30].

#### 3.2.2. Influence on Consistency (Navikow)—Research Findings

The analysis of the influence of oil number and the observations of the study indicates that, at low BCA contents, fly ash improves the consistency and workability of the mixture, acting as a filler that reduces the segregation of the components and increases the homogeneity of the structure. In this respect, this material facilitates the placement and compaction of concrete, which can have a positive effect on its mechanical properties [31].

However, at higher BCA contents (above 15–20%), the increasing water demand due to the higher specific surface area of the ash leads to excessive viscosity of the mix (Table 3) High viscosity can make processing more difficult, and, at the same time, the increased amount of batch water weakens the internal structure of the concrete, resulting in a deterioration of the strength parameters. As a result, despite the improvement in homogeneity, excess fly ash can adversely affect the mechanical properties of the mix, especially if not compensated for by appropriate proportions of superplasticizers [31].

#### 3.2.3. Adhesion to EPS (Polystyrene Foam)—Requirement ≥0.08 MPa

As the BCA content increases, a decrease in adhesion to EPS is observed, particularly at higher additive concentrations. This is due to increased porosity and higher water demand, leading to weaker interaction between the cement and the polystyrene substrate [32] (Table 3). In contrast, WL, when used in moderate amounts, improves workability and adhesion, but when used in excess, leads to a reduction in strength, due to excessive batch water [33] (Table 4).

Compared to the reference sample, formula F6 has the highest adhesion to EPS (0.12 MPa), while formula F12 has the lowest (0.06 MPa (Table 4). Formulations F1, F2 (Table 3), F5, F6, F7, F8, and F9 (Table 4) are found to meet the quality standards for adhesive concretes (EN 998-1:2016) [19]. To keep the produced concrete within the range of standards (ASTM C926) [34], the proportion of BCA should not exceed 10 percent by weight of cement, as a significant decrease in adhesion is observed above this value.

#### 3.2.4. Adhesion to Concrete Substrate—Requirement ≥ 0.25 MPa

As the BCA content of the mix increases, a decrease in adhesion to concrete is observed, particularly for formulations with a high additive content. This is due to an increased water requirement and a reduced amount of free cement particles capable of forming strong adhesive bonds with the substrate [35] (Table 3). Additionally, too much WL decreases adhesion, as excess polymers can form a layer that weakens the cement–substrate bond [36] (Table 4).

Formulations F1 (Table 3) and F5 (Table 4) achieve the highest value for adhesion to concrete (0.3 MPa), while F4 has the lowest value (0.12 MPa) (Table 3), which means that this formulation may not be suitable for structural applications requiring good adhesion to concrete. Formulations F1, F2 (Table 3), F5, F6, F7, F8, and F9 (Table 4) comply with the standards (EN 998-1:2016-12) [19], while for mixtures with higher ash content, it is recommended to compensate for the weakened adhesion by using superplasticizers [36].

The reduced adhesion observed in formulations F3 and F4 (Table 3), particularly on EPS and concrete substrates, is likely attributed to increased water demand and higher porosity associated with higher BCA contents. Excessive water not only dilutes the binder matrix but also contributes to capillary pore formation, leading to a weakened contact zone between the mortar and the substrate. In addition, the irregular particle shape and lower packing density of BCA particles can disrupt the homogeneity of the matrix, further degrading adhesion [35,36]. To improve performance, these formulations could be optimized by reducing the BCA content to ≤10%, introducing a small dosage of re-dispersible polymer powder to enhance bonding, or incorporating superplasticizers to maintain workability while lowering the water-to-binder ratio. These adjustments may enhance both the mechanical integrity and surface adhesion of the mortars, particularly in thermally insulated systems.

#### 3.2.5. Flexural Strength—Declared Value

As the BCA content of the mix increases, a reduction in flexural strength is noted, particularly above 15 percent by weight of cement. This is due to the weakening of the cementitious structure by excessive porosity, as well as a possible lower ability to compensate for stresses in the concrete [37] (Table 3). In contrast, WL, when used in moderate amounts, does not adversely affect flexural strength, but its excess can lead to a weakening of the cementitious structure through increased water retention [38] (Table 4).

Compared to the reference sample, the highest flexural strength value is achieved by formulas F5 and F6 (6.4 MPa) (Table 4), while the lowest value is achieved by formula F4 (4.6 MPa) (Table 3). The other formulations are found to meet the requirements of EN 1015-11:2001+A1:2007 [19,39] for cement mortars. To maintain the optimum strength parameters (ACI 318-19), the proportion of BCA should not exceed 12% and WL no more than 5%, as above these values, there is a significant reduction in flexural strength.

#### 3.2.6. Compressive Strength—Requirement ≥20 MPa

As the BCA content of the mix increases, a gradual decrease in compressive strength is observed, especially for large amounts of the additive. This is because fly ash does not participate in the hydration reaction of the cement to the same extent as clinker, and additionally, it increases the porosity of the concrete [40] (Table 3). WL, on the other hand, when used in appropriate proportions, improves workability, but in excess, it can weaken the cementitious structure through increased water retention and possible particle swelling [38] (Table 4).

Compared to the reference sample, Formula F5 (Table 4) achieves the highest compressive strength (26.1 MPa), and F4 (Table 3) achieves the lowest (20.5 MPa), which, nevertheless, meets the standard requirements. Formulations F1 (Table 3), F5, F6, F7, and F8 (Table 4) show the best mechanical performance and comply with EN 1015-11:2001+A1:2007 [19,39] standards. To comply with the strength standards (ASTM C39) [41], the proportion of BCA should not exceed 15%, and that of WL should not exceed 7%, as above these values, there is a significant reduction in compressive strength.

### 3.3. Synergy of the Addition of Fly Ash from Biomass Combustion and Lignin to Produce a Mortar That Meets the Requirements of European Standards

Based on the results, the synergies of BCA and WL on the properties of the cement mix were analyzed using linear regression statistical analysis (Figure 2).

For the regression analysis results, the synergy coefficients (regression slopes) indicate how BCA and WL affect key properties [42], and the *p*-values confirm the statistical significance of each effect.
BCA effects: BCA has a negative effect on all properties (except for a slight positive effect on EPS adhesion). It has a strong negative effect on compressive strength (−2.29 × 10^−9^), indicating that increasing BCA leads to a significant decrease in strength [42].WL effects: WL positively affects adhesion to EPS and concrete. It improves adhesion more effectively than BCA. It negatively affects flexural and compressive strength when it exceeds 1%, probably due to over-saturation [42].

The following interpretations can be made:BCA reduces strength and adhesion as its content increases.WL improves adhesion, but must be limited to 0.5–1% to prevent a decrease in strength.

The optimum synergy range is BCA ≤ 10% and WL between 0.5 and 1%, providing a balance between adhesion and mechanical properties.

## 4. Environmental Assessment of Cement Mortars

Samples with results adhering to EAD 040083-00-0404 [17] and EN 998-1:2016-12 [19] were selected for further environmental analysis. A reference sample (F0), samples containing BCA (F1, F2), and samples containing WL (F5, F6, F7) were selected.

### 4.1. Contaminant Leachability

Chemical analyses were performed on the leachability levels of heavy metals and other substances in the waste samples. The tests included metals such as mercury, molybdenum, nickel, lead, antimony, selenium, zinc, chlorides, fluorides, barium, sulfates, dissolved organic carbon (DOC), and dissolved solids (TDS).

According to the regulations [20], limits are in place for the following substances for waste permitted for reuse, including as raw material in the cement industry:Heavy metals (mercury, lead, cadmium, nickel, copper, zinc, selenium, molybdenum, antimony);Sulfates;Chlorides;Dissolved salts (TDS);Organic pollutants (DOC).

The following is a graphic record of contaminant leachability tests for cement mortars containing waste additives. Samples for which the obtained test results adhered to EAD 040083-00-0404 [17] and EN 998-1:2016-12 [19] standards were chosen for pollutant leaching tests.

#### 4.1.1. Contaminant Leachability of Mortars Containing BCA

The use of industrial waste as a substitute for natural raw materials in the cement industry is gaining increasing attention, both for its environmental benefits and economic feasibility. However, such materials must meet strict chemical safety and performance criteria before being incorporated into cementitious products. This article evaluates three samples (F0, F1, F2) based on the concentrations of selected heavy metals and elements, comparing them with regulatory thresholds relevant to cement use (Figure 3).

Trace Metals Below Technologically Acceptable Limits (Figure 3)

The analysis confirms that mercury (Hg), molybdenum (Mo), nickel (Ni), lead (Pb), antimony (Sb), selenium (Se), and barium (Ba) were all present at a level of leachability well below their regulatory limits:Mercury (Hg): detected at 0.01 mg/kg in all samples—right at the limit, but not exceeding it.Mo, Ni, Pb, Sb: detected at 0.04–0.1 mg/kg, which is significantly lower than their respective thresholds.Selenium (Se): varied from 0.051 to 0.075 mg/kg (limit: 0.1 mg/kg).Barium (Ba): ranged between 8.6 and 11 mg/kg, below the 20 mg/kg limit.

These findings indicate no technological or environmental barriers to using these samples in cementitious formulations [4,5,40].

2.Zinc (Zn)—Exceedance in Sample F0

Sample F0 leached 4.7 mg/kg of zinc, exceeding the acceptable limit of 4.0 mg/kg.The leaching from samples F1 (3.7 mg/kg) and F2 (2.3 mg/kg) was within the permissible range.

Implications:Excessive zinc can interfere with cement hydration, delaying setting and potentially compromising early strength development.It may also increase the porosity of the cement matrix, reducing long-term durability and increasing susceptibility to leaching.

Corrective options:Blend sample F0 with F1 and F2 to dilute the zinc concentration to an acceptable level.Alternatively, restrict its use to non-structural applications, such as backfill materials or roadbed stabilizers.

3.Conclusions and Recommendations

Samples F1 and F2 show strong potential as supplementary cementitious materials in cement production, especially as partial replacements for mineral additives.Sample F0, while chemically similar, requires zinc mitigation before full integration.Regular chemical monitoring and blending strategies are recommended to ensure regulatory compliance and performance consistency.

The reuse of such waste materials aligns with circular economy principles, reduces the extraction of virgin raw materials, and contributes to lower carbon emissions in the cement sector [4,5,40].

In addition to heavy metal analysis, the leachability of selected anions and dissolved organic components was evaluated in samples F0, F1, and F2 (Figure 4), specifically targeting chlorides (Cl^−^), fluorides (F^−^), sulfates (SO_4_^2−^), dissolved organic carbon (DOC), and total dissolved solids (TDS). These parameters are critical in assessing the potential impact of alternative raw materials on cement hydration processes, reinforcement durability, and long-term stability [4,5].

Chloride leaching in all samples (243–278 mg/kg) was well below the regulatory threshold of 800 mg/kg, indicating minimal risk of corrosion induction in reinforced concrete. However, fluoride levels significantly exceeded the limit of 10 mg/kg in all samples, ranging from 26 to 28 mg/kg. High fluoride leachability is known to retard cement setting and can disrupt the formation of hydration products, such as calcium silicate hydrates and ettringite.

Sulfate leachability was also notably above the permissible limit (1000 mg/kg), with values between 3680 and 4400 mg/kg. Elevated sulfate levels are associated with the risk of expansive reactions, particularly delayed ettringite formation (DEF), which can compromise the durability of hardened cementitious materials. While DOC values remained low (20 mg/kg across all samples), indicating minimal organic load, the TDS values were extremely high, all exceeding 35.000 mg/kg against a 4000 mg/kg limit. High TDS can negatively impact setting time, shrinkage behavior, and the ionic equilibrium within the cement matrix, especially in combination with other reactive species [4,5].

Overall, while the chloride and DOC leachability levels do not pose direct concerns, the elevated fluoride, sulfate, and TDS leachability may impair cement performance and necessitate pre-treatment or blending strategies before incorporation into binder systems. Further study on leaching behavior and hydration kinetics is recommended to confirm the suitability of these waste streams for structural applications.

#### 4.1.2. Contaminant Leachability of Mortars Containing WL

The following section presents the results of the leachability analysis of contaminants in cement mortars containing varying amounts of sodium lignosulphonate (WL) (Figure 5 and Figure 6). This analysis provides insights into the potential environmental impact and user safety associated with the use of WL as an additive.

To further validate the safety and suitability of lignosulfonate-based waste materials as alternative raw components in cementitious systems, samples F0 and F5 through F7 were assessed. The analysis focused on environmentally critical elements, including mercury (Hg), molybdenum (Mo), nickel (Ni), lead (Pb), antimony (Sb), selenium (Se), zinc (Zn), and barium (Ba) (Figure 5).

All samples demonstrated full compliance with regulatory limits for Hg, Mo, Ni, Pb, and Sb. Selenium leachability ranged from 0.046 to 0.075 mg/kg, remaining below the 0.1 mg/kg threshold, indicating minimal toxicological risk and no interference with cement hydration.

The only exceedance was noted in sample F0, where zinc reached 4.7 mg/kg, slightly surpassing the regulatory limit of 4.0 mg/kg. Elevated zinc may disrupt cement setting and increase leaching potential. In contrast, samples F5–F7 remained within safe limits, with F7 exhibiting the lowest zinc content (2.5 mg/kg).

Barium leachability was consistently low, ranging from 3.8 to 11 mg/kg—well below the 20 mg/kg limit—suggesting negligible mobility risk for Ba^2+^ ions in the cement matrix.

The results support the classification of samples F5–F7 as chemically suitable for cement product integration. Sample F0, due to its elevated zinc level, may require dilution, blending, or selective use. These findings confirm the potential for the safe reuse of lignosulfonate residues as sustainable mineral additives, contributing to waste valorization and reduced environmental impact in cement manufacturing [5].

An expanded analysis of anionic species and dissolved constituents (Figure 6) in waste-derived samples F0 and F5 through F7 reveals considerable variation in compliance with regulatory thresholds. The evaluated parameters—chlorides (Cl^−^), fluorides (F^−^), sulfates (SO_4_^2−^), dissolved organic carbon (DOC), and total dissolved solids (TDS)—serve as critical indicators of potential chemical reactivity and environmental impact when considering these materials as alternative raw inputs for cement production.

Chloride leachability was low across all samples (198–278 mg/kg), remaining well below the 800 mg/kg limit, indicating negligible risk of reinforcing steel corrosion in cementitious systems [4,5].

In contrast, fluoride leachability exceeded the permissible 10 mg/kg threshold in samples F0 (28 mg/kg), F5 (25 mg/kg), and F6 (15 mg/kg). Only sample F7 (6.5 mg/kg) fell within the allowable range. Excessive fluoride levels are known to interfere with cement hydration, potentially delaying the formation of calcium silicate hydrates and ettringite.

Sulfate concentrations were notably above the 1000 mg/kg limit in all cases, with the highest level recorded in sample F7 (6810 mg/kg). Such leachability is associated with the risk of delayed ettringite formation (DEF), which can lead to expansion and long-term degradation of hardened cement matrices [5].

DOC levels remained stable at 20 mg/kg across all samples, well below the 500 mg/kg guideline, indicating low levels of soluble organic compounds and minimal risk of biological or secondary chemical reactions.

The TDS values, however, were substantially elevated in every sample, ranging from 23,800 mg/kg (F7) to 36,200 mg/kg (F0)—all far exceeding the 4000 mg/kg limit. High TDS suggests a cumulative ionic load, which may affect setting behavior, durability, and chemical stability in cement formulations.

From an environmental and application perspective, the findings offer a mixed suitability profile:

Sample F7 emerges as the most environmentally compliant, meeting limits for both fluoride and chloride, and exhibiting the lowest sulfate and TDS levels, though still exceeding the thresholds for these.

Samples F0, F5, and F6 exceed the allowable leachability for fluoride, sulfates, and TDS, and would require pre-treatment, blending, or controlled application strategies before use in cement production.

Despite these exceedances, the samples retain potential for non-structural or blended cement applications, particularly where mechanical performance requirements are lower. Their reuse aligns with circular economy principles by enabling waste valorization and reducing dependence on virgin raw materials [4,5].

Chemical analysis of eight samples (F0–F7) showed that most met the regulatory requirements for heavy metals, such as mercury, molybdenum, nickel, lead, antimony, selenium, and barium. Zinc (Zn) exceeded the permissible level only in sample F0, suggesting the need for early dilution or restriction of use.

Anion leachability and dissolved parameters proved to be a more challenging area. All samples had exceedances for fluoride, sulfate, and TDS, with F7 showing the most favorable profile among the materials analyzed—it was the only one within the acceptable range for fluoride, and also showed the lowest leachability of TDS and sulfate. Chloride and DOC contents complied with regulations in all cases, which positively affects the potential binding properties and durability of cement [4,5]. It is worth mentioning that in most cases, the addition of waste decreased the leachability of contaminants, showing that the cement itself may be a source of contaminant leaching.

F1–F7 can be used as supplementary cementitious additives, especially after mixing or correction. F0 requires corrective action due to excess zinc content.

The excessive leachability of fluoride, sulfate, and TDS, in most samples, can be corrected through dilution, chemical stabilization, or selective application (e.g., in non-structural products).

These materials have real potential in sustainable cement production, fitting in with the circular economy and CO_2_ reduction goals.

### 4.2. Results of VOC Testing

Chromatographic analysis using mass spectrometry (GC-MS) was carried out to assess the emissions of volatile organic compounds (VOCs).

#### 4.2.1. VOC Analysis for the Reference Sample

The reference sample, F0 (labeled as a BW_01 in the chromatogram), was analyzed in detail to identify and quantify the emitted compounds.

The chromatogram obtained from the TIC (Total Ion Chromatogram) analysis shows the presence of numerous peaks, of which the most significant in terms of signal intensity are assigned to specific compounds. The main chromatographic peaks occur in the range of retention times of 6.6–38.7 min.

The most intense signals (Figure 7) were observed for the following:Cyclohexasiloxane, dodecamethyl-(RT: 18.97–19.09 min)—a compound from the siloxane group, often used in industrial products.Cycloheptasiloxane, tetradecamethyl-(RT: 23.38–23.49 min)—another representative of the same group of compounds.Cyclooctasiloxane, hexadecamethyl-(RT: 26.73–26.84 min)—known for its wide use in the chemical and cosmetic industries.Cyclopentasiloxane, decamethyl-(RT: 14.14–14.22 min)—a typical component of industrial silicones.

Based on the NIST library [25] and spectral mass analysis, the following compounds were identified with a high match to the reference library (>50% match probability):Hydrazinecarboxamide (CAS: 57-56-7)—a compound used in chemical synthesis processes (Figure 8).Acetic acid, hydrazide (CAS: 1068-57-1)—a potential by-product of chemical reactions (Figure 9).5-Amino-2-methyl-2H-tetrazoles (CAS: 6154-04-7)—a compound used in the pharmaceutical and energy industries (Figure 10).

The presence of cyclic siloxanes, such as octamethylcyclotetrasiloxane (D4) and decamethylcyclopentasiloxane (D5), is notable, as these compounds are commonly used in personal care products (Table 5), but have raised increasing concern in the context of indoor air pollution. D4 and D5 are classified as persistent, bioaccumulative, and toxic (PBT) substances under certain environmental frameworks [43]. Prolonged inhalation exposure to siloxanes has been associated with potential endocrine disruption and neurotoxicity, although their specific thresholds in building materials are still under review. Likewise, 2.5-Hexanedione, a diketone and known neurotoxic metabolite of n-hexane, has been linked to chronic nervous system effects in occupational environments. While no specific indoor regulatory limit currently exists for these compounds in construction products, their detection in WL-rich formulations indicates a need for caution in enclosed or poorly ventilated spaces. These findings underscore the importance of VOC monitoring in the adoption of new bio-based admixtures, especially when considering their implementation in thermal insulation systems and interior applications [23].

#### 4.2.2. Comparison with Samples Containing BCA

Samples F1–F2 contained different amounts of BCA fly ash, which affected the chemical composition of the substances analyzed.
F1 (average quantity of fly ash): The main compound detected was 2,5-Hexanedione (C_6_H_10_O_2_), with high intensity, suggesting increased presence of this compound compared to in sample F0.F2 (increased amount of fly ash): Increased amounts and intensities of hydrocarbon compounds and organic nitrogenous compounds were observed. The peak at 19.034 min (high intensity) indicates an increased content of volatile organic compounds associated with the ash.

The Total Ion Chromatogram (TIC) (Figure 11) analysis shows the intensity of the ion signal as a function of retention time and serves as an indicator of the total volatile organic compound (VOC) emissions from the samples. Samples F0, F1, and F2 show a similar profile, with the main emission peak located around 20 min of retention time, suggesting the presence of a dominant VOC fraction of medium volatility [44].

Sample F2 generated the highest total TIC intensity, peaking at around 2200 units, indicating the highest volatile organic content in this sample.

Sample F1 shows a moderate intensity, and Sample F0 shows the lowest level of VOC emissions among the samples analyzed.

The decrease in intensity after the peak is gradual and parallel in all samples, suggesting a similar composition of functional groups, differing only in quantity.

Relevance to cement applications:

A high VOC content can adversely affect air quality at the application site, especially in indoor applications.

Samples with lower emissions (F0, F1) are more desirable from an environmental and health standpoint, and may also carry a lower risk of secondary reactions in the cement matrix.

The F2 sample, despite its potential chemical advantages, may require additional purification or reduction of organic fractions if it is to be used in low-emission systems [44].

Significant differences in the amount of organic matter were detected (Figure 12), including an increase in some aromatic compounds and high-molecular-weight hydrocarbons.

The main differences between the reference sample and the fly ash samples were as follows:

The more fly ash in the sample, the higher the peak intensity for some organics, especially in the 18–22 min retention time range.

New substances appeared in samples F1–F2 that were not present in sample F0.
-2.5-Hexanedione appeared in samples with fly ash (F1–F3), suggesting its presence as a product of combustion or interaction with ash.-Nitrogenous and nitro-organic compounds (e.g., nitrobenzene, tetrazoles) were only present in samples F2 and F3, which may suggest their formation by chemical reactions in the presence of more fly ash.-Boronic acid, diethyl-, and other boron compounds may originate from fly ash impurities or catalysts used in the combustion process.-Methylfurfural and Hydroxyacetone indicate the presence of biomass degradation products or organic residues in the sample.

The increase in 2.5-Hexanedione and other oxidation products indicates possible chemical reactions occurring during combustion.

This comparison highlights the relative intensity of volatile organic compound (VOC) emissions for two specific retention time peaks. The peak at 19.034 min represents a major VOC component, while the 14.184 min peak may correspond to a lighter, more volatile compound.

Sample F2 shows the highest intensity for both peaks, suggesting it contains the greatest quantity of VOCs among the three.

Sample F0 consistently shows the lowest VOC emission profile, reinforcing previous conclusions from TIC data [44].

The consistent increase in both peaks from F0 → F1 → F2 indicates a trend of rising organic content and possibly more complex or heavier VOC constituents in higher-numbered samples.

Samples with higher peak intensities may pose a greater environmental burden in terms of air emissions and odor.

F0 is the most suitable for applications where low VOC emissions are critical, such as indoor materials or low-emission types of cement.

Sample F2 may require pre-treatment or usage limitations, due to its higher VOC content.

The relative peak areas (Figure 13) provide insight into the dominant volatile organic compounds (VOCs) emitted from each sample. The chart illustrates the percentage contribution of five identified compounds to the total VOC profile for each sample.
Sample F2 consistently exhibits the highest relative peak areas across all compounds, indicating a higher VOC load and greater chemical diversity.The compound contributing the most in all samples is the unidentified peak at 19.034 min, reaching 40% in F2, followed by 35% in F1, and 30% in F0. This suggests a major shared VOC constituent with increasing intensity across the series.2.5-Hexanedione, a potentially neurotoxic compound, also increases in abundance from F0 (10%) to F2 (15%), raising concerns about occupational exposure depending on application.Other oxygenated organics, like hydroxyacetone and methylfurfural, also follow the increasing trend, which may influence odor and environmental interactions.

Sample F0 demonstrates the lowest relative VOC content, reinforcing its suitability for low-emission cementitious applications.

Sample F2, while rich in potentially reactive VOCs, may still be used with ventilation or mitigation strategies or restricted to non-sensitive applications.

The consistency in compound types across samples suggests similar decomposition patterns or precursor content, with differences primarily in concentration [44].

The chart illustrates a clear upward trend in VOC intensities from sample F0 to F2 (Figure 14). All measured substances, including 2.5-hexanedione, hydroxyacetone, methylfurfural, and nitrobenzene, show increasing values, indicating a progressive enrichment in volatile organic compounds.
The peak at 19.034 min consistently shows the highest intensity, confirming its dominant role in the VOC profile.Sample F2 exhibits the highest VOC emission, while F0 remains the cleanest in terms of volatile output.

F0 is the most favorable for low-emission applications, while F2 may require treatment or usage in ventilated environments, due to its higher VOC content.

An increase in BCA content leads to increased emissions of certain volatile organic compounds, which may be relevant both for chemical composition analysis and for assessing the environmental impact of co-firing. In particular, the formation of compounds such as nitrobenzene may indicate potentially harmful chemical effects resulting from the presence of fly ash [44].

#### 4.2.3. Comparison with Samples Containing WL

GC-MS analyses for samples F0 (reference sample) and samples F4–F7 show differences in the composition of volatile compounds that can be related to the content of sodium lignosulphonate (WL) (Figure 15).

Samples with sodium lignosulphonate (F4–F7) show differences in the intensity and presence of specific volatile compounds. For example, samples F4–F7 show the presence of 2,5-Hexanedione and 3-Hexene-2,5-dione, which are not as strongly marked in the reference sample.
The reference sample (F0) has a lower signal intensity compared to samples with sodium lignosulfonate.Samples F5–F7 show stronger and more pronounced peaks between 20 and 30 min, indicating the increased presence of volatile compounds.The highest intensity is recorded for F7, suggesting a higher concentration of specific compounds.Peak shifts across samples suggest that sodium lignosulfonate influences the composition and dynamics of volatile compound release.

The appearance of new compounds in the lignosulphonate samples, such as siloxanes (Cyclooctasiloxane, Cyclononasiloxane), may indicate interactions between the lignosulphonate and other sample components.

Changes in the intensity of the chromatograms can be observed. Sample F7 shows a stronger signal for peaks at RTs of approximately 19,040 min and 21,325 min, suggesting the presence of compounds characteristic of higher lignosulphonate content.

Analysis of the total ionograms (TICs) clearly shows an increase in signal intensity in samples containing sodium lignosulfonate (F5–F7) compared to the reference sample (F0). The greatest differences are observed in the retention time interval of 20–30 min, suggesting a more intense release of volatile compounds with the addition of WL (Figure 15). Specifically, sample F7 shows the highest signal values, which may indicate a saturation or accumulation effect of volatile compounds with a higher proportion of WL. The observed peak shifts suggest changes in the dynamics of chemical reactions occurring during cement hydration with the presence of WL [23].

Cyclononasiloxane and Cyclooctasiloxane show a higher presence in samples F5 and F7 relative to F0, indicating that the WL content influences their concentration.

2.5-Hexanedione and 3-Hexene-2.5-dione have a notable presence in samples F5 and F6, suggesting their relevance in lignosulfonate interactions.

The reference sample, F0, has a minimal amount of these compounds, confirming that WL significantly alters the volatile profile.

Data on the relative peak areas of key compounds (Figure 16), such as Cyclonasiloxane and Cyclooctasiloxane, show a marked increase in F5 and F7 samples relative to the reference sample. These compounds are typical by-products of reactions involving organic substances in alkaline environments. Additionally, the presence of diketone (2.5-Hexanedione, 3-Hexene-2.5-dione) suggests that WL may partially degrade or react with other cement mortar components, generating new VOC emissions.

Cyclononasiloxane and Cyclooctasiloxane show a steady increase in intensity with WL content, peaking for sample F7 (Figure 17).

2.5-Hexanedione and 3-Hexene-2.5-dione exhibit fluctuations, with peaks for samples F5 and F6, suggesting possible interactions with WL.

The reference sample, F0, has the lowest compound intensities, reinforcing the impact of WL on volatile compound composition.

Silicone compounds and diketones show different trends depending on WL content. Siloxanes increase steadily with the amount of additive (F5 → F7), confirming their strong relationship with lignosulfonate content. Diketones, on the other hand, show non-linear trends, with a maximum for samples F5 and F6, which may indicate a complex mechanism of formation, dependent on the WL concentration and its interaction with cement and water. Such profiles suggest that the influence of WL is not only quantitative, but also qualitative, affecting the type and quantity of the compounds formed [23].

Sodium lignosulfonate has a clear effect on the chemical profile of the samples, increasing the abundance of specific volatile compounds, such as siloxanes and diketones. These findings suggest that WL influences not only the chemical nature, but also the dynamic release and concentration of volatile compounds, which can be crucial for understanding its role in various applications [45].

#### 4.2.4. Summary of VOC Tests

This study analyzed the emissions of volatile organic compounds (VOCs) for three types of cement mortar samples:A reference sample (without additives);Samples containing BCA;Samples containing sodium lignosulphonate (WL).

The baseline sample showed the lowest VOC emissions, which was expected, as it contained no additional organic components.

The components of the standard cement mortar did not generate significant amounts of VOCs.

The presence of BCA increased emissions of some VOCs compared to the reference sample. Possible sources of emissions were the organic residues present in the ash and its interaction with cement and water [46]. However, the increased emissions were not drastic, suggesting that BCA can be used as an additive in cement mortars, but its impact on indoor air quality requires further study.

The samples containing sodium lignosulphonate (WL) showed significantly higher VOC emissions compared to both the reference sample and the BCA sample. Lignosulphonates, as by-products of the cellulose industry, may contain organic compounds with potential volatility that are released during cement hydration [47]. It is also possible that they enter into chemical reactions in the cement mixture, leading to the formation of additional volatile by-products [47].

The addition of BCA may have a moderate effect on VOC emissions, suggesting the possibility of its use in cement mortars, with monitoring of environmental aspects [23,43].

Sodium lignosulphonate (WL), despite its plasticizing properties, significantly increases VOC emissions, which may limit its use in some conditions, especially in confined building spaces [43].

Although WL acts as a plasticizer, its impact on VOC emissions may limit the applicability of this additive in confined building spaces. In the context of sustainable construction, further investigation of lignosulfonate modifications or searching for alternative additives that do not generate such high levels of VOC emissions is needed. Equally important is a long-term assessment of the impact of these additives on air quality and the health of building occupants [48].

### 4.3. Results of the Carbon Footprint Calculations and Life Cycle Assessment (LCA) for the Samples with BCA and WL

#### 4.3.1. Carbon Footprint Calculations

Carbon footprint calculations were carried out according to the established methodology described in earlier chapters, based on the ISO 14040 and ISO 14044 Life Cycle Assessment (LCA) guidelines. A reference sample and formulations containing waste raw materials, including fly ash from biomass co-firing and sodium lignosulphonate, a lignin derivative, were analyzed.

The calculations aimed to determine the impact of using substitutes for virgin raw materials on the carbon footprint of the final product. The results are summarized below (Table 6).

The bar chart (Figure 18) illustrates the annual carbon footprint (expressed in megagrams of CO_2_ per year) for the reference mortar formulation and selected alternative formulations (F0, F1, F2, F5–F7). The analysis was conducted within a cradle-to-gate scope, covering emissions associated with raw material extraction, production, and supply [25].

The reference formulation (F0) exhibits the highest carbon footprint, exceeding 1440 Mg CO_2_/year, due to the exclusive use of virgin, energy-intensive materials. Samples F1 and F2, which incorporate increasing amounts of biochar-based additives (BCA), show only minor reductions (on the order of 0.5–2.0%), indicating limited mitigation potential at low substitution levels [25].

Formulations F5 to F7, modified by reducing emission-intensive additives (e.g., cellulose ether and EVA copolymer), demonstrate more substantial improvements. In particular, sample F7—containing 0.25% sodium lignosulfonate (WL)—achieves the lowest footprint, at approximately 1300 Mg CO_2_/year, representing a reduction of about 10% compared to the reference.

These results highlight that while low-percentage BCA substitution offers minimal environmental benefit, replacing synthetic additives with renewable or waste-derived alternatives (such as WL) can lead to more significant emission reductions.

#### 4.3.2. LCA Analysis

The Life Cycle Assessment (LCA) was conducted within a cradle-to-gate scope (Modules A1–A3), following the general principles of ISO 14040/44 [26,27] and aligned with the requirements of EN 15804 [28] for construction products.

The analysis included the following:Raw material extraction and processing (A1);Transportation of raw materials to the production site (A2);Product manufacturing (A3).

The results are summarized below (Table 7).

The bar chart (Figure 19) presents the annual carbon footprint (Modules A1–A3) for the reference formulation and the cementitious mixtures of samples F1–F7. The analysis reflects emissions generated from the production and supply of raw materials (cradle-to-gate).

The reference recipe emits over 1509 Mg CO_2_/year.

Samples F1–F2 introduce increasing amounts of biomass ash, but the impact on carbon reduction is minimal.

Samples F5–F6 reduce the content of carbon-intensive additives, like cellulose ether and EVA copolymer, achieving a modest reduction in emissions (~3–3.4%).

Sample F7, which incorporates WL (sodium lignosulfonate), shows the most significant reduction of all (without biochar), reaching around 1300 Mg CO_2_/year—roughly 14% lower than the reference.

Figure 19 systematizes the impact of the analyzed options from a cradle-to-gate perspective. The largest contributors to total emissions are the extraction and processing stages of raw materials. The reduction of chemical additives (F5–F6) and their partial replacement with WL lignosulfonate (F7) leads to a noticeable reduction in emissions throughout the cycle analyzed by LCA. Sample F7 not only has the lowest overall carbon footprint, but also the greatest reduction in unit emissions per ingredient in the formulation [49].

A carbon footprint analysis was carried out for 11 samples of cement mortar formulations, including a reference sample and mixtures containing waste raw materials (ash, sodium lignosulphonate, biochar). The calculations were performed in the A1–A3 (cradle-to-gate) range, according to the LCA methodology and EN 15804 [28].

While the LCA results demonstrate that formulations with waste lignin (WL) exhibit a lower environmental footprint, particularly in terms of CO_2_ emissions and embodied energy, it is important to acknowledge the technical limitations associated with higher WL contents. As shown in earlier sections, formulations containing WL above 0.5% tend to exhibit reduced mechanical strength, which may restrict their use in load-bearing or high-stress applications [49]. This trade-off between ecological gain and mechanical compromise underscores the importance of defining optimal formulation ranges—in this case, WL contents of 0.1–0.2% were found to offer the best balance between environmental benefits and structural adequacy. Such balanced formulations can be confidently proposed for non-structural applications, including thermal insulation adhesives in ETICS systems, where reduced VOC emissions and improved workability are equally critical. Future development of hybrid systems or functional additives may further expand this window of performance [49].

The use of waste raw materials and the reduction of carbon-intensive additives significantly reduce the carbon footprint of cement mortars. Even small changes in composition can have a measurable environmental effect, making these solutions attractive in the context of decarbonizing construction.

## 5. Limitations and Future Perspectives

Despite the promising outcomes regarding the use of biomass combustion ash (BCA) and sodium lignosulphonate (WL) as additives in cement mortars, several limitations were identified that must be acknowledged for practical application and future research.

### 5.1. Limitations of Biomass Combustion Ash (BCA)

Water demand and workability: Increasing the BCA content beyond 10% significantly raised water demand, due to its fine particle size and high specific surface area. While it acts as a filler at lower concentrations, higher amounts can adversely affect the workability and flow of the mortar [50].

Adhesion and porosity: Mortars with >15% BCA showed reduced adhesion to substrates (EPS and concrete) and increased porosity, compromising mechanical integrity.

Material variability: The composition of BCA can vary depending on biomass type and combustion conditions, potentially affecting reproducibility and performance consistency across different production batches [45].

### 5.2. Limitations of Sodium Lignosulphonate (WL)

VOC emissions: One of the most critical limitations of WL is its significant contribution to the emission of volatile organic compounds (VOCs). As observed through GC-MS analysis, increasing the WL content (especially >0.5%) markedly elevated the presence of siloxanes and diketones, raising concerns for indoor air quality and regulatory compliance [51].

Mechanical properties: While WL improves plasticity and workability, excessive use results in decreased compressive and flexural strength, due to water retention and interference with cement hydration [51].

Impurities and stability: Being a by-product of the pulp and paper industry, WL may contain residual organics or impurities that can degrade or react in alkaline cementitious environments, leading to unpredictable behavior [51].

### 5.3. Future Perspectives

To overcome the outlined limitations, several research directions should be pursued.

Standardization of BCA composition: Developing guidelines for the classification and preparation of BCA will enhance its reliability in industrial applications [52].

Purification and chemical modification of WL: Refining WL to reduce VOC precursors could make it more viable for indoor applications [52].

Combined additive optimization: Investigating synergistic formulations with BCA, WL, and other industrial by-products may yield mortars with balanced mechanical, rheological, and environmental profiles.

Life cycle and long-term studies: Future work should include long-term performance studies, life cycle toxicity, and durability in different climatic and usage conditions.

This study provides a strong foundation for the sustainable use of BCA and WL in building materials, but also highlights the importance of addressing technical and environmental challenges to support their widespread adoption in construction.

## 6. Conclusions

This study confirms that lignocellulosic industrial by-products—biomass combustion ash (BCA) and waste lignin (WL)—can serve as functional and environmentally conscious additives in cement mortars. A summary of the data is included in Table 8. The addition of BCA (1–10%) improved mortar homogeneity, adhesion to both EPS and mineral substrates, and workability. WL (0.1–max 0.5%) functioned effectively as a natural plasticizer, enhancing viscosity and application efficiency.

Mechanically, both additives contributed positively to selected performance parameters, particularly when dosed within optimal ranges. Environmentally, BCA and WL align well with the principles of circular economy, enabling the reuse of renewable biomass-derived residues and reducing reliance on virgin raw materials. Life Cycle Assessment (LCA) confirmed a reduced carbon footprint, especially in WL-modified formulations.

Importantly, leachability tests revealed that the contaminant concentrations in all BCA- and WL-based mortars were well below regulatory limits, indicating a favorable environmental safety profile under standard conditions. However, VOC analyses indicated the presence of compounds such as nitrobenzene and cyclic siloxanes in mortars with a higher WL content. Although the total emissions were moderate, these findings suggest that environmental risks, while low, are not absent, particularly in enclosed or poorly ventilated settings. As a precaution, responsible dosage and adequate ventilation are recommended when applying WL-rich formulations.

To contextualize these results, a comparative summary of BCA and WL against conventional additives was developed. This comparison highlights their potential and limitations in terms of function, sustainability, and technical performance.

In conclusion, cement mortars incorporating BCA and WL represent a promising step toward more sustainable construction materials. These additives contribute meaningfully to performance optimization, resource efficiency, and emission reduction goals, while being compatible with modern regulatory and ecological frameworks. Nevertheless, trade-offs between environmental and mechanical performance must be carefully managed by:Future research should aim to do the following:Refine dosage strategies for optimal performance;Investigate VOC mitigation techniques (particularly for WL);Conduct long-term durability tests;Validate performance in real-world field applications;Explore synergistic formulations with other circular additives.

Such efforts will support the safe and effective implementation of these materials in modern, environmentally responsible construction.

## Figures and Tables

**Figure 1 materials-18-02086-f001:**
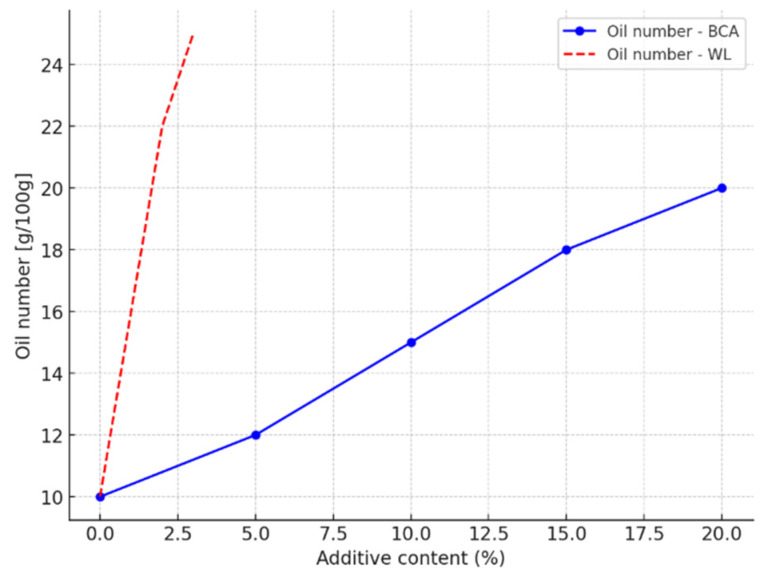
Plot of dependence of oil number on BCA and WL content in mortars.

**Figure 2 materials-18-02086-f002:**
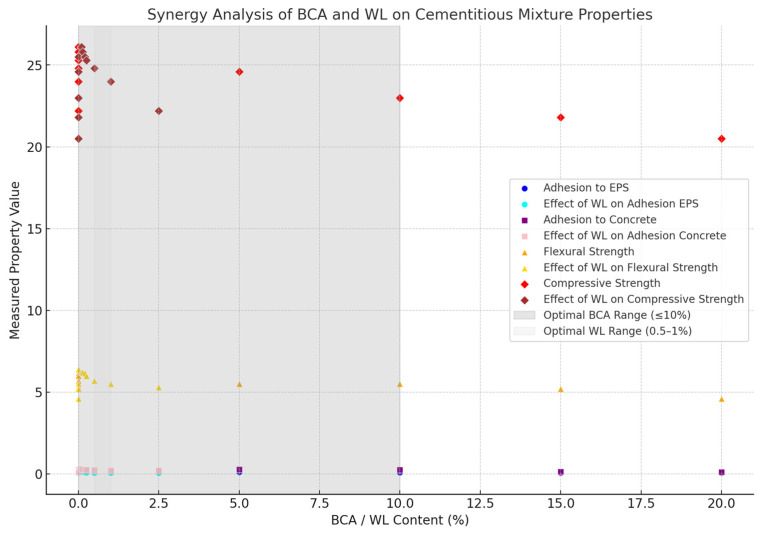
Plot of regression analysis results for BCA and WL.

**Figure 3 materials-18-02086-f003:**
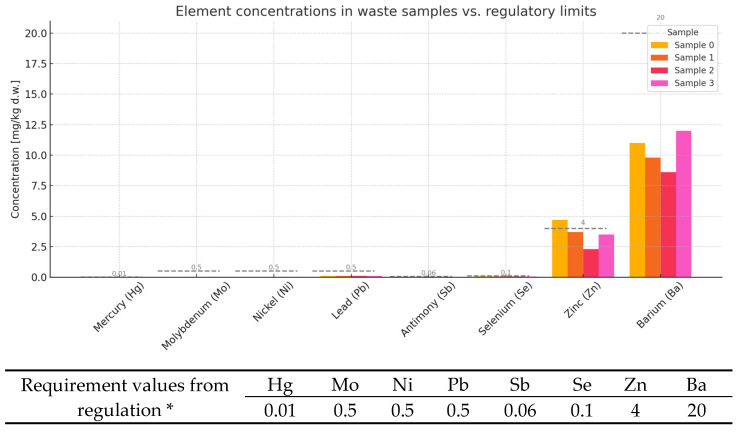
Plot of leachability of elements depending on BCA content in mortars. * Degree of leaching of elements (mg/kg) from mortars containing BCA (reference formula, F0, compared to formulas F1–F2) and comparison with threshold values from Regulation of the Minister of Economy of 16 July 2015 [20] on acceptance of inert waste at landfills, Annex No. 2.

**Figure 4 materials-18-02086-f004:**
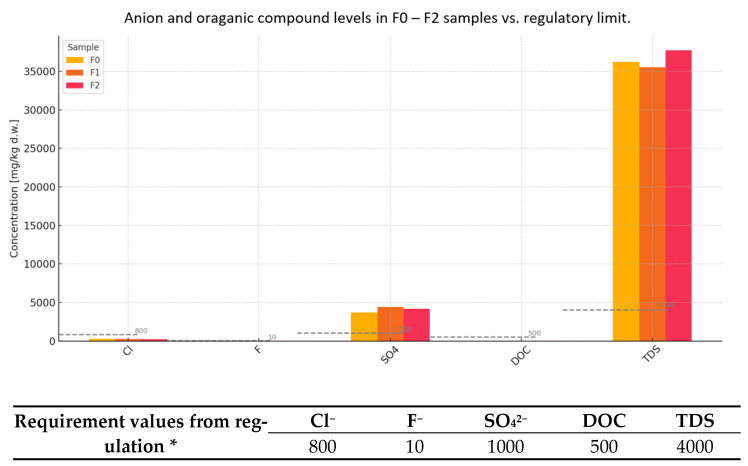
Plot of leachability of elements depending on BCA content in mortars (sulfates, chlorides). * Degree of leaching of elements (mg/kg) from mortars containing BCA (reference formula, F0, compared to formulas F1–F3) and comparison with threshold values from Regulation of Minister of Economy of 16 July 2015 [20] on acceptance of inert waste at landfills, Annex No. 2.

**Figure 5 materials-18-02086-f005:**
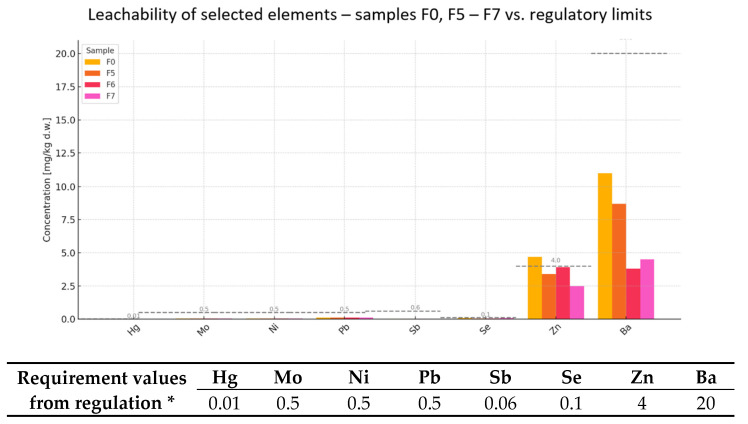
Plot of leachability of elements depending on WL content in mortars. * Degree of leaching of elements (mg/kg) from mortars containing WL (reference formula, F0, compared to formulas F4–F7) and comparison with threshold values from Regulation of the Minister of Economy of 16 July 2015 [20] on acceptance of inert waste at landfills, Annex No. 2.

**Figure 6 materials-18-02086-f006:**
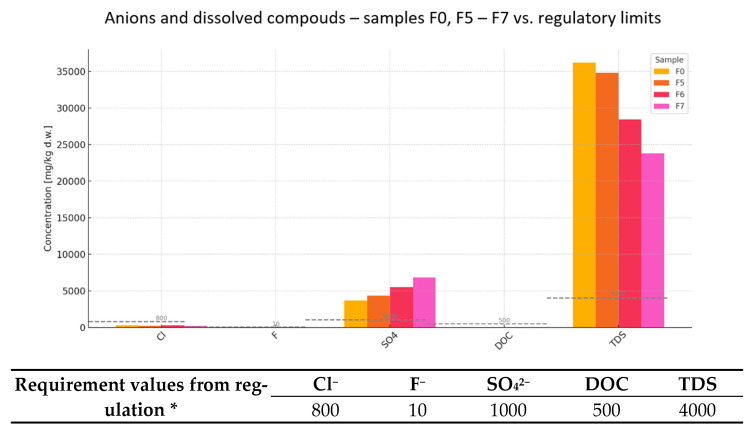
The Plot of pollutants’ leachability depends on WL content in mortars (sulfates, chlorides). * Degree of leaching of elements (mg/kg) from mortars containing WL (reference formula, F0, compared to formulas F4–F7) and comparison with threshold values from Regulation of the Minister of Economy of 16 July 2015 [20] on acceptance of inert waste at landfills, Annex No. 2.

**Figure 7 materials-18-02086-f007:**
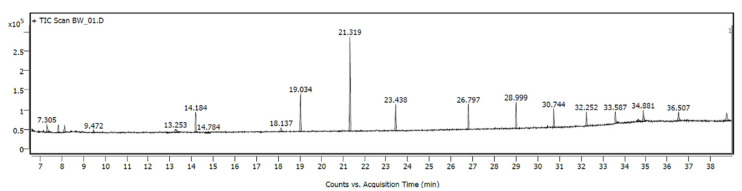
The chromatogram obtained from the TIC (Total Ion Chromatogram) for the reference sample, F0.

**Figure 8 materials-18-02086-f008:**
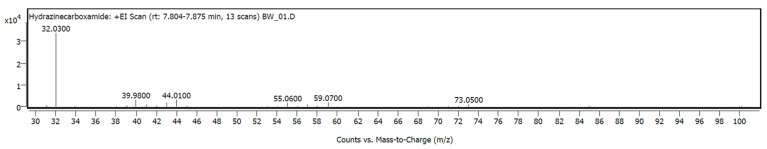
The chromatogram of Hydrazinecarboxamide content in the reference sample.

**Figure 9 materials-18-02086-f009:**
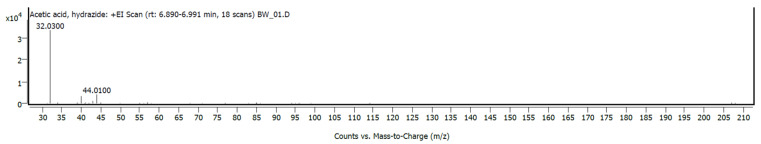
The chromatogram of acetic acid content in the reference sample.

**Figure 10 materials-18-02086-f010:**
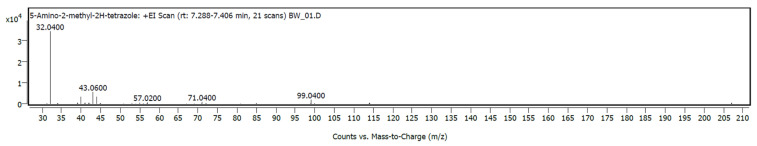
The chromatogram of 5-Amino-2-methyl-2H-tetrazole content in the reference sample.

**Figure 11 materials-18-02086-f011:**
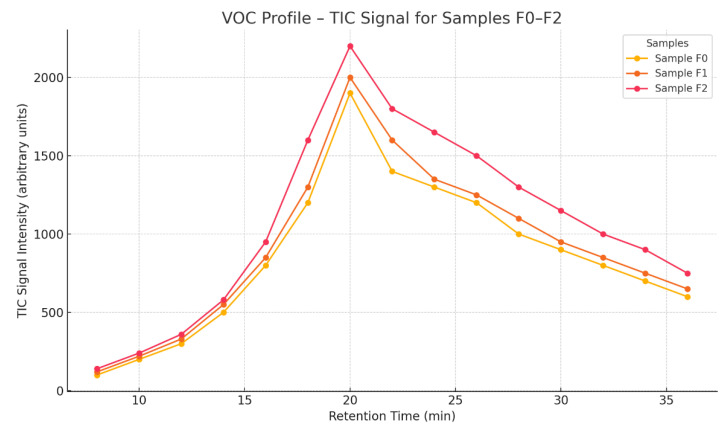
Total Ion Chromatogram (TIC) comparison for samples F0 and F1–F3.

**Figure 12 materials-18-02086-f012:**
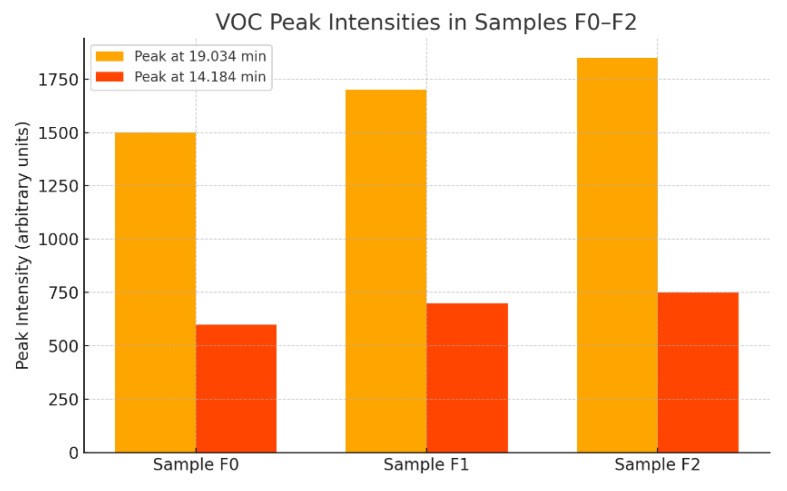
Comparison of peak intensities at 19.034 min and 14.184 min.

**Figure 13 materials-18-02086-f013:**
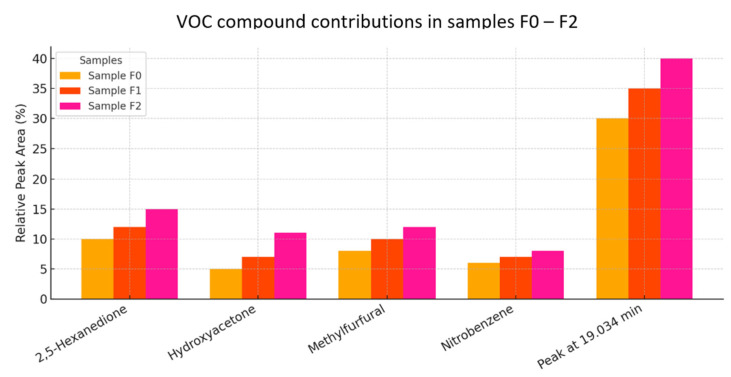
Comparison of relative peak areas of major volatile compounds in samples F0–F3.

**Figure 14 materials-18-02086-f014:**
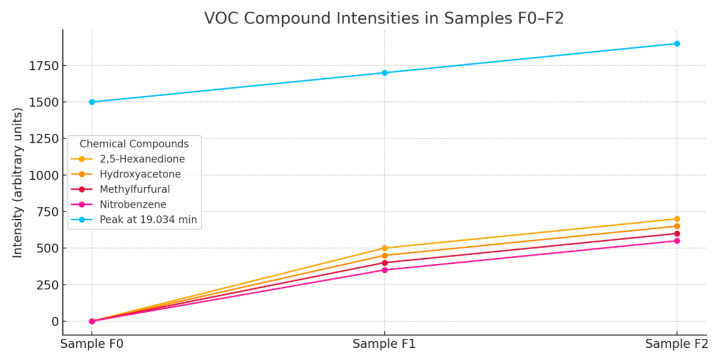
Changes in compound intensities across samples.

**Figure 15 materials-18-02086-f015:**
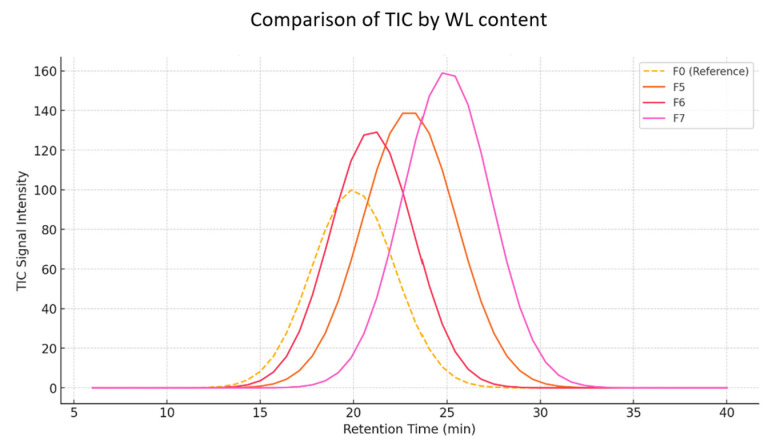
Comparison of TICs by WL content.

**Figure 16 materials-18-02086-f016:**
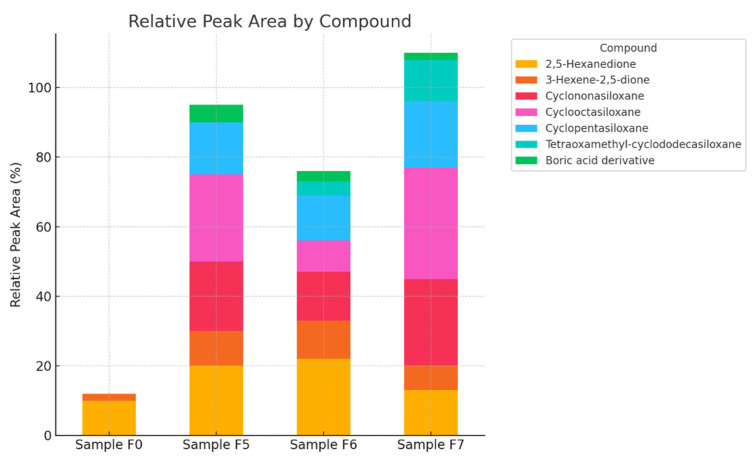
Comparison of relative peak areas of key volatile compounds.

**Figure 17 materials-18-02086-f017:**
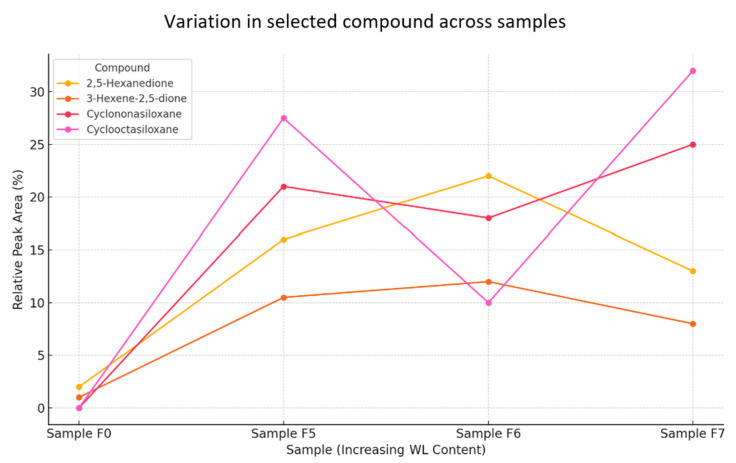
Trends of selected compounds with WL content.

**Figure 18 materials-18-02086-f018:**
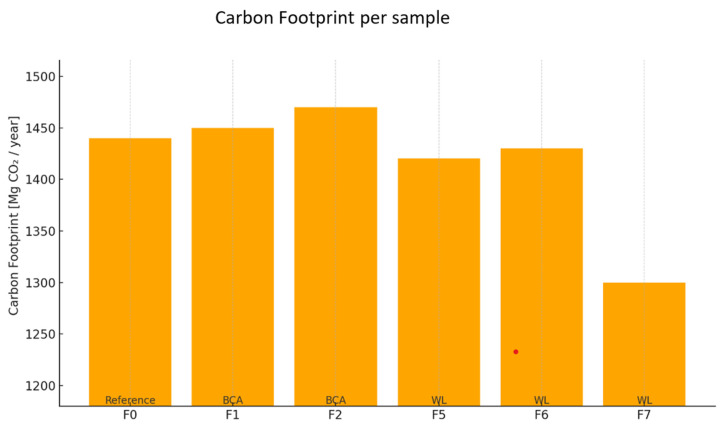
Comparison of annual carbon footprint for reference sample vs. samples F1–F7. All formulations maintain the same cement content (30%) while containing varying amounts of other components, such as quartz sand, additives, and waste-based materials (biomass ash, biochar, sodium lignosulfonate).

**Figure 19 materials-18-02086-f019:**
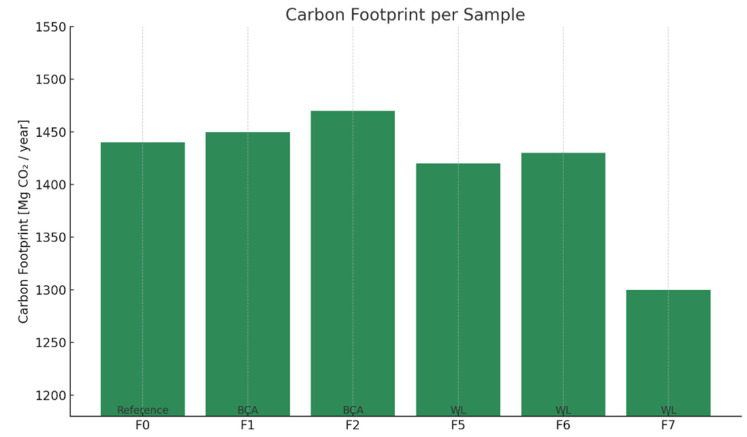
Annual carbon footprint (LCA modules).

**Table 1 materials-18-02086-t001:** Percentage composition of mortar with different additions of BCA waste materials.

Ingredients	Reference Formula	F1	F2	F3	F4
	[%]	[%]	[%]	[%]	[%]
Cementitious mixture	100	95	90	85	80
BCA	0	5	10	15	20
Total:	100	100	100	100	100

**Table 2 materials-18-02086-t002:** Percentage composition of mortar with different additions of WL waste materials.

Ingredients	Reference Formula	F5	F6	F7	F8	F9	F10	F11	F12
	[%]	[%]	[%]	[%]	[%]	[%]	[%]	[%]	[%]
Cementitious mixture	100	99.9	99.87	99.8	99.75	99.5	99	97.5	96
WL	0	0.1	0.12	0.2	0.25	0.5	1	2.5	4
Total:	100	100	100	100	100	100	100	100	100

**Table 4 materials-18-02086-t004:** Averaged test results for formulations developed with the addition of WL content. Bolded numbers show which formulations meet the standards for mortar.

Ingredient	F5	F6	F7	F8	F9	F10	F11	F12	Requirement [17,19]
**Water–cement ratio (by oil number)**	0.21	0.215	0.22	0.23	0.24	0.24	0.25	0.25	
**Consistency** **(Navikow)**	**6.5**	**7**	**7**	**7**	**7**	**7**	**7.5**	8	6.5–7.5
**Adhesion to EPS**	**0.11**	**0.12**	**0.09**	**0.08**	**0.08**	0.07	0.65	0.06	≥0.08 MPa
**Adhesion to concrete substrate**	**0.3**	**0.29**	**0.28**	0.24	0.22	0.21	0.2	0.19	≥0.25 MPa
**Adhesion to EPS at +5 °C (EPS)**	**0.1**	**0.11**	**0.09**	**0.08**	**0.08**	0.07	0.06	0.05	≥0.08 MPa
**Adhesion to concrete substrate +5 °C (EPS)**	**0.28**	**0.25**	0.22	0.2	0.2	0.2	0.18	0.17	≥0.25 MPa
**Flexural strength**	6.2	6.2	6.15	6	5.7	5.5	5.3	5.2	Declared value
**Compressive strength**	**26.1**	**25.8**	**25.5**	**25.3**	**24.8**	**24.0**	**22.2**	**21.7**	(≥20 MPa) EN 1015-11:2001+A1:2007

**Table 5 materials-18-02086-t005:** Identified chemical compounds and match probabilities.

Compound	CAS Number	Match Probability (%)
Cyclohexasiloxane, dodecamethyl-	540-97-6	45.41
Cycloheptasiloxane, tetradecamethyl-	107-50-6	64.81
Cyclooctasiloxane, hexadecamethyl-	556-67-2	52.77
Hydrazinecarboxamide	57-56-7	45.41
Acetic acid, hydrazide	1068-57-1	29.58
5-Amino-2-methyl-2H-tetrazole	6154-04-7	45.41

**Table 6 materials-18-02086-t006:** Annual carbon footprint for formulations developed.

	Variant	Carbon Footprint [kg CO_2_/100 kg]	Carbon Footprint [Mg CO_2_/Year]
(reference)	F0	27.23	1509.48
BCA	F1	27.23	1509.65
	F2	27.24	1510.31
WL			
	F5	26.25	1455.51
	F6	26.09	1446.50
	F7	23.40	1297.53

**Table 7 materials-18-02086-t007:** Annual LCA for formulations developed.

	Variant	Carbon Footprint [Mg CO_2_/Year]
(reference)	F0	1509.48
BCA	F1	1509.65
	F2	1510.31
WL	F5	1455.51
	F6	1446.50
	F7	1297.53

**Table 8 materials-18-02086-t008:** Comparative summary of functions, advantages, and limitations of cement mortar additives.

Additive	Function/Effect	Advantages	Drawbacks
Biomass combustion ash (BCA)	Pozzolanic/filler	Reuses waste; moderate strength gain; potential CO_2_ reduction	High water demand; porosity; variable composition
Sodium lignosulphonate (WL)	Plasticizer/dispersant	Improves workability; renewable origin	High VOC emissions; strength reduction at high content
Fly ash	Pozzolanic	Improves durability and strength; widely studied	May affect early strength; regional availability
Metakaolin	Pozzolanic/reactive	Enhances early strength; improves chemical resistance	Cost; limited availability
Silica fume	Pozzolanic/densifier	Reduces permeability; increases strength	Workability issues; requires superplasticizers

## Data Availability

The original contributions presented in this study are included in the article. Further inquiries can be directed to the corresponding author.

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
