# Peer review of "The Effects of Biomass Combustion Ash and Lignin on the Properties of Cement Mortars and Their Environmental Impact"

_materials, 2025, doi:10.3390/ma18092086_

Round 1
Reviewer 1 Report
Comments and Suggestions for Authors
This manuscript presents the use of wastes as functional additives in cement mortars, which contributes to the circular economy and sustainable construction. Biomass combustion ash (BCA) and waste lignin (WL) were used as additives in cement mortars. These additives improved the durability and energy efficiency in cementitious systems. The authors can enhance their manuscript based on the following comments:
1.-In the first paragraph of the Introduction section should improve the description of the research problem and challenges. In addition, this manuscript has many short paragraphs. These paragraphs should be rewritten. Also, this section should enhance the description of the advantages and limitations of the other additives used in cement mortars. Finally, the last paragraph should consider more novelty and advantages of the proposed research.
2.- Line 142, the words "3. Results " must be corrected.
3.- Table 1 should be improved.
4.- The sub-section 3.2 has many short paragraphs. These paragraphs should be improved.
5.-The resolution and technical quality of the Figures 3, 4, 5, and 6 should be enhanced.
6.-The authors should include more discussions of the behavior of the results reported in the Figures 15, 16, 17, 18, and 19.
7.- The sub-sections 3.4 and 3.5 have many short paragraphs. These paragraphs should be rewritten.
8.- The authors should consider discussions of the limitations of the proposed additives for cement mortars.
9.- The authors should incorporate a Table with the main parameters, advantages, and drawbacks of the proposed additives for cement mortars compared to other additives reported in the literature.
10.- What are the future research works?
11.- The conclusions must be enhanced based on the previous comments.
12.- The authors should add more recent references between 2024 and 2025.
13.- The format of references must be improved.
Comments on the Quality of English Language
English grammar and style of all the sections of the manuscript must be enhanced.
Reviewer 2 Report
Comments and Suggestions for Authors
Dear author, I attach a document with my comments. Regards!

Reviewer 3 Report
Comments and Suggestions for Authors
The manuscript presents a comprehensive study on the use of biomass combustion ash (BCA) and waste lignin (WL) as additives in cement mortars, aligning with circular economy principles. The research is well-structured, with clear objectives and robust methodologies. However, several areas require revision to enhance clarity, scientific rigor, and readability.
- Abstract (Page 1): The abstract is concise but lacks specific quantitative results to highlight the study’s impact. For instance, it mentions improved durability and energy efficiency but does not quantify the extent of improvement (e.g., percentage increase in adhesion or reduction in carbon footprint). Revise to include key numerical findings, such as the optimal BCA (1-10%) and WL (0.5-1%) ranges and their specific effects on mortar properties, to strengthen the abstract’s impact.
- Introduction (Pages 1-2): The introduction effectively contextualizes the use of BCA and WL but could benefit from a clearer justification of why these specific waste materials were chosen over other alternatives (e.g., fly ash from coal or other industrial by-products). Add a brief comparison with other waste materials to clarify the novelty and relevance of BCA and WL in sustainable construction.
- Materials and Methods (Pages 3-6, Section 2): The description of BCA and WL preparation is detailed, but the rationale for selecting specific percentages (e.g., BCA from 5-20%, WL from 0-4%) is unclear. Provide a brief explanation of how these ranges were determined, possibly referencing preliminary studies or literature, to justify the experimental design.
- Results (Page 7, Section 3.1): The oil number results (Figure 1) are well-presented, but the discussion lacks depth regarding the practical implications of increased water demand due to higher BCA and WL content. Expand this section to discuss how increased water requirements might affect construction practices, such as mixing time or curing processes, and suggest potential mitigation strategies (e.g., superplasticizers).
- Results (Page 8, Section 3.2): The adhesion results (Table 2) show that some formulations (e.g., F3, F4) fail to meet the standard requirements for adhesion to EPS and concrete. However, the manuscript does not discuss why these formulations underperformed or propose adjustments to improve their performance. Add a paragraph analyzing the reasons for reduced adhesion (e.g., increased porosity or water demand) and suggest modifications to optimize these formulations.
- Results (Pages 10-12, Section 3.3): The contaminant leachability results are robust, but the absence of molybdenum and nickel data for WL samples (noted on Page 12) is a significant gap. Address this by either providing the missing data or explaining why these elements were not analyzed, along with the potential implications for environmental safety assessments.
- Results (Pages 12-17, Section 3.4): The VOC analysis is thorough, but the discussion of siloxanes and other compounds (e.g., 2,5-Hexanedione) lacks context on their environmental and health impacts. Revise to include a brief explanation of the significance of these compounds in construction materials, referencing relevant literature on their toxicity or regulatory limits, to enhance the environmental relevance of the findings.
- Results (Pages 19-20, Section 3.5): The LCA results (Table 5, Table 6) demonstrate a reduced carbon footprint for formulations with WL, but the manuscript does not discuss the trade-offs between environmental benefits and mechanical performance (e.g., reduced strength in high-WL formulations). Add a discussion on balancing environmental and technical performance to guide practical applications of these mortars.
- Summary (Page 21, Section 4): The summary effectively highlights the study’s contributions but overstates the environmental safety of the mortars by claiming they pose “no environmental risk” (Page 12). Given the presence of VOCs like nitrobenzene (Page 15), moderate this claim to reflect that risks are minimal but not entirely absent under standard conditions. Additionally, suggest future research directions, such as long-term durability tests or field applications, to strengthen the conclusion.
Round 2
Reviewer 1 Report
Comments and Suggestions for Authors
The authors addressed the reviewer's comments.